# TIME SERIES SUBSEQUENCE ANOMALY DETECTION VIA GRAPH NEURAL NETWORKS

## ABSTRACT

Time series subsequence anomaly detection is an important task in a large variety of real-world applications ranging from health monitoring to AIOps, and is challenging due to complicated underlying temporal dynamics and unpredictable anomalous patterns. Firstly, how to effectively learn the temporal dependency in time series remains a challenge. Secondly, diverse and complicated anomalous subsequences as well as the lack of labels make accurate detection difficult. For example, the popular subsequence anomaly detection algorithm—time series discord—fails to handle recurring anomalies. Thirdly, many existing algorithms require a proper subsequence length for effective detection, which is difficult or impossible in practice. In this paper, we present a novel approach to subsequence anomaly detection which combines practical heuristics of time series discords and temporal relationships with deep neural networks. By performing length selection considering multi-scale information and incorporating prior knowledge using graph neural networks, our method can adaptively learn the appropriate subsequence length as well as integrated representations from both priors and raw data favorable to anomaly detection. In particular, our graph incorporates both semantic and temporal relationships between subsequences. The experimental results demonstrate the effectiveness of the proposed algorithm, which achieves superior performance on multiple time series anomaly benchmarks in comparison with state-of-the-art algorithms. Codes and datasets are available online[1].

## 1 INTRODUCTION

Detecting anomalies in time series data has a large variety of practical applications, such as tracing patients' bio-signals for disease detection (Chauhan & Vig, 2015), monitoring operational data of cloud infrastructure for malfunction location (Zhang et al., 2021), finding risks in IoT sensing time series (Cook et al., 2019), etc. It has received a great amount of research interests (Keogh et al., 2005; Yankov et al., 2007; Boniol & Palpanas; Shen et al., 2020; Lu et al., 2022).

The time series anomaly detection (TSAD) problem is commonly formulated to locate anomalies at each point of the time series (namely point-wise TSAD). However, this formulation fails in considering temporal relationships of anomalous points as anomalies can go beyond occurring point by point but tend to exist consecutively over a time interval in many real-world scenarios. For instance, some demand patterns from the power system change during holidays. Figure 1 shows a comparison of point-wise anomalies and subsequence anomalies. In this paper, we investigate TSAD from a subsequence perspective by identifying anomalous patterns in a time interval, which is called time series subsequence anomaly detection. Generally speaking, a subsequence anomaly is a sequence of observations that deviates considerably from some concept of normality. The somewhat "vague" definition itself also hints the challenges of the subsequence anomaly detection problem. Also, a distinguishing feature of time series is temporal dependency. Thus, how to learn and utilize the temporal dependency for different time series data is a key challenge in time series anomaly detection. Moreover, another key challenge in time series subsequence anomaly detection is how to determine the appropriate subsequence length, as illustrated in Figure 2. This problem becomes worse when there are multiple abnormal subsequences with different lengths in one series.

---

[1] https://anonymous.4open.science/r/GraphSAD-B082

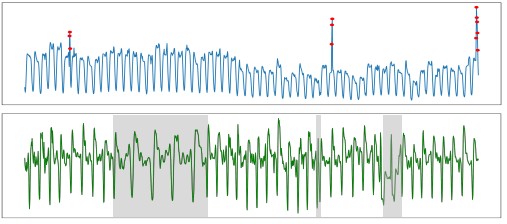

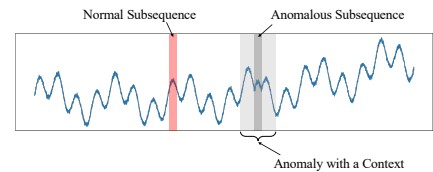

Figure 1: Point-wise Anomalies (Top) versus Subsequence Anomalies (Bottom). The top is a website traffic time series with anomalies labeled by red dots that might be caused by cyberattacks. The bottom is an insect's activity signal recorded with an EPG apparatus, where time intervals marked in grey are subsequences exhibiting different anomalous characteristics, including period length variation, spike, and temporal morphological change.

Figure 2: An Illustration of Selecting Proper Length for Subsequence Anomaly Detection. An anomalous subsequence is inside the dark grey zone. If we directly detect anomalies using this length, the anomaly might not be found as it is very similar to normal subsequences, e.g., the green zone. Instead, it is better to select a longer window size (marked in light grey) including the anomaly with its context to highlight the anomalous pattern.

Early research on anomaly detection mainly relies on shallow representations Breunig et al. (2000); Schölkopf et al. (2001); Tax & Duin (2004). Later, Deep-SVDD (Ruff et al., 2018) enhances the representation learning capability using neural networks. Recently, TSAD methods (Carmona et al., 2022; Shen et al., 2020) based on Deep-SVDD are prevailing due to their excellent performance. They introduce a suitable neural architecture for modeling time series and detecting anomalies by computing the distance between a target with its corresponding reference in latent representation space, where the reference represents normal patterns. The main issue is that these deep learning-based methods are difficult to enforce assumptions of anomalies, and typically require large training datasets to learn accurate models. In contrast, time series discord (Keogh et al., 2005; Yeh et al., 2016; Nakamura et al., 2020; Lu et al., 2022) is another category of distance-based TSAD methods. Discords are subsequences that are maximally different from all others in the time series, where the difference is measured via z-normalized Euclidean distance. The most appealing merit of discords is that anomalies can be discovered by merely examining the test data without a training phase. In spite (or perhaps because) of their extremely simple assumptions, discords are competitive with deep learning methods. However, there are still several important limitations that prevent them from broader applications. First, they fail in detecting anomalous patterns recurring at least twice, as each occurrence will be the others' nearest neighbor. Second, they rely on an explicit distance measure (z-normalized Euclidean distance), which cannot account for diversified anomalies flexibly, as some anomalous patterns might be slight in the data space. Details about the Deep-SVDD and discord algorithms are provided in Appendix B.

Moreover, most existing methods utilize a predefined window length to detect anomalies, which is difficult or impossible to tune in practice. Here we emphasize the importance of the appropriate window size which highlights the normal or anomalous pattern of a subsequence. On the one hand, the duration of anomalies varies. For example, if we use a large window to detect spikes and dips, the anomalies might be "hidden" in the normal data. While for a long-term anomaly, a short window cannot depict the full picture of it. On the other hand, even if we have a prior anomaly length, it is still necessary to intelligently infer a suitable length according to data characteristics. To the best of our knowledge, none of the existing algorithms can detect anomalous subsequences with different lengths and characteristics intelligently. For more details of related work, an extensive literature review is provided in Appendix A.

In this paper, we devote to resolve the aforementioned challenges in TSAD by fusing practical heuristics of time series discords with deep neural networks, and propose GraphSAD, a graph neural network-based subsequence anomaly detection approach. Specifically, we construct graphs in which nodes represent subsequences and edges encode the relationship between corresponding subsequences. A unique feature of our graph is that we consider both the pair-wise semantic and temporal relationships between subsequences. As a result, the temporal dependency from time series is incorporated in graph and utilized in further detection. Besides, in order to intelligently learn subsequence length, we introduce a multi-scale feature encoder to generate representations of multi-scale subsequence length with a length selection strategy to select the proper length. The proposed algorithm GraphSAD can intelligently detect different anomalous subsequences, which greatly im-

proves its usages in practice. When evaluated on multiple TSAD benchmark datasets, GraphSAD consistently outperforms state-of-the-art baselines.

## 2 PRELIMINARIES

### 2.1 NOTATIONS AND PROBLEM DEFINITION

A univariate time series of length $T$ can be denoted as $\boldsymbol{x} = (x_1, x_2, \cdots, x_T)$, where $x_t$ is a real-valued observation at time step $t$. A subsequence $\boldsymbol{x}_{t,L} = (x_t, x_{t+1}, \cdots, x_{t+L-1})$ is a series of continuous observations from time series $\boldsymbol{x}$ starting at time step $t$ with length $L$, where $1 \leq t \leq T - L + 1$. The problem of subsequence anomaly detection is to output the anomaly score $\mathrm{s}(\boldsymbol{x}_{t,L})$ for each subsequence which should be high for anomalous data and low for the normal. Then one can sort the scores in descending order to detect anomalies.

Using the sliding window strategy with a stride $\tau$, a time series can be split into an ordered set of subsequences $\mathbf{X} \in \mathbb{R}^{N \times L}$, where $N$ is the number of subsequences and $L$ is the subsequence length. We denote the $i$-th subsequence $\boldsymbol{x}_{(i-1)\tau+1,L}$ as $\mathbf{X}_i \in \mathbb{R}^L$. We assume that the initial subsequence length $L$ is set to be large enough to support detecting various anomalies, which can be typically determined based on domain knowledge. For instance, $L$ can be four times of the period length for some periodic time series, e.g., the weather time series.

Note that in most real-world applications, finding and labeling anomalies is extremely time-consuming and expensive, while the labeled normal data are much easier to access. Hence, TSAD methods are usually trained with normal and unlabeled data which are commonly treated as normal data as we assume that most data are normal due to the rareness of anomalies.

### 2.2 GRAPH NEURAL NETWORKS

Graph neural networks ( Wu et al. (2021), GNN) operate on a graph $\mathcal{G} = (V, A, \mathbf{E})$, where $V = [N] := \{1, \cdots, N\}$ is the set of node indices, $A \in \{0, 1\}^{N \times N}$ is the adjacency matrix with $A_{ij} = 1$ denoting a directed edge $(i, j)$ exists and $A_{ij} = 0$ otherwise, and $\mathbf{E} \in \mathbb{R}^{N \times N \times d_e}$ is a sparse tensor of edge attributes and $d_e$ is the corresponding dimension. The set of adjacent neighbors of $i$ is notated as $\mathcal{N}_i = \{j \in \mathcal{V} | A_{ji} = 1\}$. Let $\mathbf{H} \in \mathbb{R}^{N \times d}$ be node representations, in which the $i$-th row $\mathbf{H}_i \in \mathbb{R}^d$ is the vectorized representation attached to node $i$. A GNN takes the node representations $\mathbf{H}$ along with the graph $\mathcal{G}$ as input and returns updated node representations $\mathbf{H}' \in \mathbb{R}^{N \times d'}$ using a message passing paradigm, i.e., $\mathbf{H}' = \mathrm{GNN}(\mathcal{G}, \mathbf{H})$. The operations for each node performed by a GNN used in this work are defined as follows:

$$\text{Message and Aggregation}: \quad \mathbf{H}_{\mathcal{N}_i} = \sum_{j \in \mathcal{N}_i} \mathrm{MLP}\left([\mathbf{H}_j; \mathbf{E}_{j,i}]\right),$$

$$\text{Update}: \quad \mathbf{H}'_i = \mathrm{MLP}\left([\mathbf{H}_i; \mathbf{H}_{\mathcal{N}_i}]\right),$$

(1)

where $\mathrm{MLP}(\cdot)$ stands for a certain multilayer perceptron network and $[\cdot; \cdot]$ denotes vector concatenation. In the remaining part of this paper, we treat $\mathrm{GNN}(\cdot)$ as a multi-layer message-passing GNN to avoid notational clutter.

## 3 METHODOLOGIES

In this section, we describe GraphSAD, a graph neural network-based subsequence anomaly detection approach. Figure 3 shows the proposed architecture. Once a time series is split into subsequences, we construct semantic and temporal graphs representing data priors. Then a multi-scale feature encoder generates subsequence representations of multi-scale lengths and a length selection mechanism selects proper subsequence length. Finally, taking subsequence representations and prior graphs as inputs, graph networks generate the final representations for anomaly detection.

### 3.1 ENCODING PRIORS AS GRAPHS

At the very beginning, we would like to incorporate informative heuristics of time series discords and temporal relationships into our approach by encoding these priors as graphs. Compared with

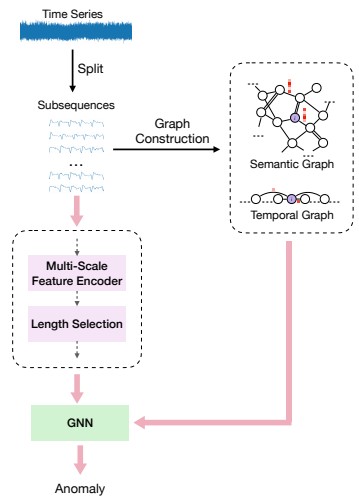

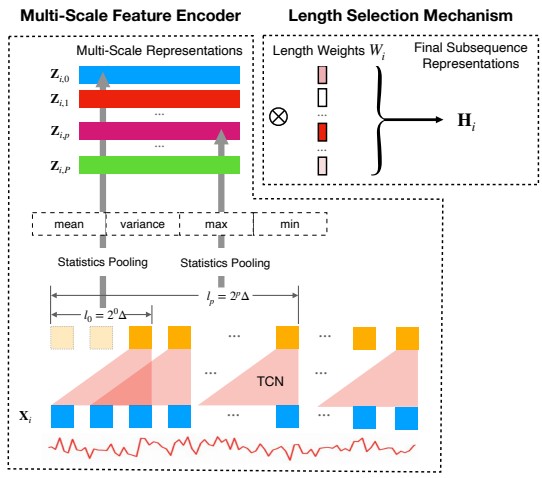

Figure 4: Learning Subsequence Representations Using Multi-Scale Encoder and Length Selection Mechanism.

Figure 3: An Overview of GraphSAD.

existing works, our method considers both the temporal information and distance information in the graph construction.

**Semantic Graph.** An attributed directed semantic graph is constructed to encode the nontrivial distance information between pair-wise subsequences, which can be described by a 3-tuple: $\mathcal{G}^{(\text{se})} = (V, A^{(\text{se})}, \mathbf{E}^{(\text{se})})$, where $V = [N]$ is the set of node indices representing a subsequence each, $A^{(\text{se})} \in \{0, 1\}^{N \times N}$ is the adjacency matrix, and edge attributes $\mathbf{E}^{(\text{se})} \in \mathbb{R}^{N \times N \times d_e}$ store the distance information between subsequences. We build the semantic graph following three principles: (1) multiple distance measures should be considered to characterize semantic relationships from different perspectives, (2) distance information of multi-scale length should be introduced, and (3) only informative neighbors of each node are reserved for the sparsity and purity of the graph.

We take the edge $(j, i)$ (a directed edge from $j$ to $i$) and the corresponding edge attribute $\mathbf{E}^{(\text{se})}_{j,i} \in \mathbb{R}^{d_e}$ as an example to illustrate how to build the semantic graph. First, we derive multi-scale views of subsequences with maximum length $L$. Specifically, the multi-scale view of subsequence $\mathbf{X}_i$ is defined as $\mathcal{X}_i = \{\mathbf{X}_{i,:l} | l = 2^p \Delta, p = 0, 1, \cdots, P\}$, where $\mathbf{X}_{i,:l}$ indicates the first $l$ observations of $\mathbf{X}_i$ with $l$ increasing exponentially, $P$ is the largest positive integer satisfying $2^P \Delta \leq L$, and $\Delta$ is the length of an indivisible segment. We calculate Euclidean distance $d_{ij,l}$ and z-normalized Euclidean distance $d^{\text{z-norm}}_{ij,l}$ between subsequences $i$ and $j$ of length $l = 2^p \Delta$, $p = 0, 1, \cdots, P$, and each distance $d_{ij}$ is then transformed to a semantic proximity $w_{ji} = \exp\left(\frac{-(d_{ij})^2}{\sigma^2}\right)$ as an entry of the edge attribute $\mathbf{E}^{(\text{se})}_{j,i}$, where $\sigma$ is the standard deviation of this distance. Moreover, the semantic graph follows the definition of the $K$-NN graph to enforce sparsity, i.e., edge $(j, i)$ is present if any distance between $i$ and $j$ is among the $K$-th smallest distances between $i$ and other nodes.

**Temporal Graph.** One distinguishing feature of time series data is its temporal dependency. To effectively encode the temporal information, we explicitly build the temporal graph $\mathcal{G}^{(\text{te})} = (V, \mathbf{A}^{(\text{te})}, \mathbf{E}^{(\text{te})})$ for time series, where $V$ is node indices consistent with the semantic graph , $\mathbf{A}^{(\text{te})}$ the adjacency matrix of temporal graph, and $\mathbf{E}^{(\text{te})} \in \mathbb{R}^{N \times N \times 1}$ denotes the edge weights for temporal proximity of subsequences. Specifically, we consider two types of edge connections in graph construction. The first one encodes the temporal dependency in the neighborhood. We connect temporally adjacent subsequences by setting the corresponding edge weight to $1$. Intuitively, we assume the neighboring subsequences are more likely to share similar dynamics. The other one captures the periodic temporal dependency for periodic time series by constructing the periodic connection. For periodic time series with period length $m^2$, the subsequences $\boldsymbol{x}_{t,l}$ and $\boldsymbol{x}_{t+m,l}$ are connected in the

---

[2]We use auto-correlation to detect whether a time series is periodic as well as the period length.

temporal graph. To deal with the possible time warping, we introduce additional connections with less weights for certain tolerance: for any neighbors of node $t + m$, i.e., $|j - (t + m)| \leq r$ for given radius $r$, we have

$$w_{t,j} = \exp(-|j - t - m - 1|), \qquad (2)$$

which stands for the periodic edge weight and is encoded in $\mathbf{E}^{(\text{te})}$. Note that the weight decays exponentially as the difference between time indices increases.

**Anomaly Injection.** As we mentioned above, the labeled anomaly data are rare for training in most situations. The anomaly injection method is used to inject several labeled artificial anomalous subsequences into the time series. Here we consider six different types of anomalies, including spike & dip, resizing, warping, noise injection, left-side right, and upside down. In more detail, spike & dip stands for standard point anomaly with unreasonable high or low value, resizing randomly changes the temporal resolution of subsequence, warping denotes random time warping and noise injection for random noise injected, while the latter two stand for reversement of subsequence from left to right and up to down, respectively. Despite the fact that such artificial anomalies may not coincide with real anomalies, it is beneficial for learning characteristic representations of normal behaviors.

## 3.2 Learning Representations of Subsequences with Multi-Scale Lengths

While selecting a proper subsequence length that highlights the characteristics of both normal and anomalous patterns is important for AD, most of the existing algorithms commonly set the subsequence length through trial and error or by some prior knowledge (e.g., the period length for a periodic time series). A potential modification is to perform anomaly detection with multiple subsequence lengths and vote for anomalies, while it is still not robust since anomaly scores of different lengths are hard to align. Thus, in order to learn expressive subsequence representations with proper length, we propose a temporal convolution network ( Bai et al. (2018), TCN) based multi-scale feature encoder aggregating information at different length resolutions as well as a learnable length selection mechanism to generate representations of proper length.

**Multi-Scale Feature Encoder.** This encoder takes the raw subsequence $\mathbf{X}_i \in \mathbb{R}^L$ as input, and output the multi-scale representations $\mathcal{Z}_i = \{\mathbf{Z}_{i,0}, \cdots, \mathbf{Z}_{i,P}\}$, where $P$ is the number of length scales, defined in Section 3.1. First, the TCN consists of several layers of causal convolution with ReLU activation and layer normalization outputs the intermediate embeddings $\mathbf{R}_i = \text{TCN}(\mathbf{X}_i) \in \mathbb{R}^{L \times d}$, where $d$ is the hidden dimension and necessary padding is performed to keep the subsequence length unchanged. Then, we aggregate information of variable length $l$ by pooling $\mathbf{R}_{i,:l} \in \mathbb{R}^{l \times d}$ into a single vector. Despite the fact that mean-pooling or max-pooling is frequently used to generate a single vector summarizing a sequence, they fail to characterize the statistical property adequately which is crucial for anomaly detection. For example, The maximum and minimum are helpful for detecting spike and dip anomalies, but the variance increases when noises occur. To integrate this inductive bias into representation space, we define statistics pooling operator $\text{StatsPool}(\cdot)$ that calculates and concatenates basic statistics including mean, variance, maximum, and minimum of an $l$-length subsequence embedding through the first dimension (time axis):

$$\mathbf{Z}_{i,p} = \text{StatsPool}(\mathbf{R}_{i,:l}) = [\text{mean}(\mathbf{R}_{i,:l}); \text{var}(\mathbf{R}_{i,:l}); \max(\mathbf{R}_{i,:l}); \min(\mathbf{R}_{i,:l})] \in \mathbb{R}^{4d}, \qquad (3)$$

where length $l = 2^p \Delta$ is the $p$-th scale length. By performing statistics pooling with multi-scale length from scale 0 to $P$, we can generate the multi-scale representations $\mathcal{Z}_i = \{\mathbf{Z}_{i,0}, \cdots, \mathbf{Z}_{i,P}\}$.

**Length Selection Mechanism.** As multi-scale subsequence representations are generated, the next problem is how to select proper lengths among them. We propose to learn a length selection embedding $\mathbf{W}_i \in \mathbb{R}^P$ for each subsequence. Then the subsequence representation $\mathbf{H}_i$ can be generated as follows:

$$\mathbf{Z}_i = \text{Softmax}(\mathbf{W}_i)(\mathbf{Z}_{i,0}, \cdots, \mathbf{Z}_{i,P})^\top \in \mathbb{R}^{4d}, \qquad (4)$$

$$\mathbf{H}_i = \text{MLP}(\mathbf{Z}_i) \in \mathbb{R}^d. \qquad (5)$$

The length selection embedding $\mathbf{W}_i$ is initialized to an all-zero vector for anomaly-agnostic scenarios, which is then jointly learned with model parameters (in Section 3.4). As a result, the weights of different lengths are the same in the beginning, and during the training procedure, this embedding is optimized to pay more attention to the proper length that minimizes the anomaly detection objective. It is worth mentioning that one can introduce priors of subsequence length by adjusting the

initialization of $\mathbf{W}_i$ to $\boldsymbol{e}^{(p)} = [0, \ldots, 1, \ldots, 0] \in \mathbb{R}^P$ with a 1 at position $p$. For instance, $\boldsymbol{e}^{(0)}$ can be set for the time series dataset in which point-wise anomalies (e.g., spike and dip) often occur.

## 3.3 Multi-Context Aware Anomaly Detection

Once subsequence representations $\mathbf{H} \in \mathbb{R}^{N \times d}$ are obtained, we enhance them with prior knowledge encoded in graphs via GNN. For a target node (subsequence), the networks utilize representations of its neighboring nodes as well as the proximity information (i.e., distance information in the semantic graph and temporal information in the temporal graph) encoded in neighboring edges and generate context-enhanced representations, which are then concatenated as the final representations:

$$\mathbf{H}^{(\mathrm{se})} = \mathrm{GNN}_1(\mathcal{G}^{(\mathrm{se})}, \mathbf{H}) \in \mathbb{R}^{N \times d}, \tag{6}$$

$$\mathbf{H}^{(\mathrm{te})} = \mathrm{GNN}_2(\mathcal{G}^{(\mathrm{te})}, \mathbf{H}) \in \mathbb{R}^{N \times d}, \tag{7}$$

$$\mathbf{H}' = \left[ \mathbf{H}^{(\mathrm{se})}; \mathbf{H}^{(\mathrm{te})} \right] \in \mathbb{R}^{N \times 2d}, \tag{8}$$

where $\mathrm{GNN}_1(\cdot)$ and $\mathrm{GNN}_2(\cdot)$ are multi-layer graph neural networks performing on the semantic graph and temporal graph, respectively.

Following the distance-based anomaly detection methods, our approach identifies anomalies by comparing the target representation with a reference. Note that in our implementation the reference considers both semantic and temporal context. Formally, we propose a distance-based anomaly score function, namely a multi-context aware hypersphere anomaly score defined as

$$s(\mathbf{X}_i) = \alpha \left\| \mathbf{H}'_i - \mathbf{C}^{(\mathrm{se})}_i \right\|^2 + (1 - \alpha) \left\| \mathbf{H}'_i - \mathbf{C}^{(\mathrm{te})}_i \right\|^2, \tag{9}$$

where $\alpha \in [0, 1]$ is a hyperparameter, $\mathbf{C}^{(\mathrm{se})}_i$ and $\mathbf{C}^{(\mathrm{te})}_i$ are the semantic and temporal context of $\mathbf{X}_i$, respectively, which are the mean of node representations within $S$-hop neighbors of $\mathbf{X}_i$ in the two graphs and can be calculated in a matrix form as

$$\mathbf{C} = \left( \mathrm{Binary} \left( \sum_{s=1}^{S} \mathbf{A}^s \right) - \mathbf{I} \right)^{\top} \mathbf{H}', \tag{10}$$

where the context $\mathbf{C} \in \mathbb{R}^{N \times 2d}$ can be either $\mathbf{C}^{(\mathrm{se})}$ or $\mathbf{C}^{(\mathrm{te})}$, and so does the adjacency matrix $\mathbf{A}$. $\mathbf{A}^s$ returns the $s$-hop adjacency matrix, where $(\mathbf{A}^s)_{i,j}$ is number of $s$-hop path from node $i$ to $j$. $\mathrm{Binary}(\cdot)$ is a binarization function applied on each entry of a matrix, which returns 1 if the entry is greater than 0 and returns 0 otherwise. As a result, the $i$-th row of $\left( \mathrm{Binary} \left( \sum_{s=0}^{S} \mathbf{A}^s \right) - \mathbf{I} \right)^{\top}$ can represent nodes that can reach $i$ within $S$ hops (node $i$ itself is removed).

## 3.4 Model Learning

By incorporating artificially labeled anomalies, we propose a loss function $\mathcal{L}$ following the Hypersphere Classifier (HSC) (Ruff et al., 2020) objective with the multi-context aware anomaly score

$$\mathcal{L} = \frac{1}{N} \sum_{i=1}^{N} -(1 - y_i) s(\mathbf{X}_i) - y_i \log(1 - \exp(-s(\mathbf{X}_i))), \tag{11}$$

where $s(\mathbf{X}_i)$ is the anomaly score of subsequence $\mathbf{X}_i$ (see Equation 9), and $y_i \in \{0, 1\}$ with 1 for (artificial) anomalous subsequences and 0 for normal and unlabeled subsequences. Details about HSC is summarized in Appendix B. Furthermore, in the scenarios with both normal and unlabeled data in the training set, we actually use a weighted version of HSC objective, where normal data is assigned with a larger weight than unlabeled data, as if a known normal sample gets a high anomaly score, it needs to be penalized more.

**Regularizations.** We introduce two regularizations for stable learning including auto-encoding regularization and Laplacian regularization.

*Auto-encoding regularization.* We introduce an auxiliary MLP decoder to reconstruct the original subsequence $\mathbf{X}_i$ using the output of GNN $\mathbf{H}'_i$. The auto-encoding regularization is defined as

$$\mathcal{L}_{\mathrm{dec}} = \frac{1}{N} \sum_{i=1}^{N} \mathrm{MSE}(\mathbf{X_i}, \mathrm{Dec}(\mathbf{H}'_i)), \tag{12}$$

where $\mathrm{MSE}(\cdot)$ denotes the mean squared error loss and $\mathrm{Dec}(\cdot)$ is the decoder. Suppose that artificial anomalies are not involved, i.e., $y_i = 0$ for all training samples, the network might corrupt to map

all samples to a constant representation and achieve the optimal. This regularization is utilized to restrict the network to preserve the information of input data to avoid trivial solutions. Even with anomaly injection, it can still conduce to the model robustness.

*Laplacian regularization of length selection embedding.* We impose a restriction that the length selection embeddings of adjacent nodes in both graphs should be close, which can be implemented with a Laplacian regularization defined as

$$\mathcal{L}_{\text{Lap}} = \sum_{\hat{\mathbf{A}}} \sum_{\substack{\forall i,j \in \mathcal{V}| \\ \hat{\mathbf{A}}_{i,j}=1}} \|\mathbf{W}_i - \mathbf{W}_j\|^2 = \sum_{\hat{\mathbf{A}}} \sum_{p=1}^{P} \mathbf{W}_{:,p}^T (\hat{\mathbf{D}} - \hat{\mathbf{A}}) \mathbf{W}_{:,p}, \tag{13}$$

where $\mathbf{W} \in \mathbb{R}^{N \times P}$, the row vector $\mathbf{W}_i \in \mathbb{R}^P, \forall i \in \{1, \cdots, N\}$ is the length selection embedding of each node, $\mathbf{W}_{:,p} \in \mathbb{R}^{N \times 1}, \forall p \in \{1, \cdots, P\}$ is the column vector, $\hat{\mathbf{A}} = \max(\mathbf{A}, \mathbf{A}^{\top})$ returns a symmetric adjacency matrix (i.e., makes graphs undirected), and $\mathbf{A} \in \{\mathbf{A}^{(\text{se})}, \mathbf{A}^{(\text{te})}\}$. $\hat{\mathbf{D}} \in \mathbb{R}^{N \times N}$ denotes the diagonal degree matrix of the undirected graph with $\hat{\mathbf{D}}_{ii} = \sum_j \hat{\mathbf{A}}_{ij}$, and $\hat{\mathbf{D}} - \hat{\mathbf{A}}$ returns the unnormalized graph Laplacian.

**Training Procedure.** As two components need to be optimized, i.e., model parameters $\boldsymbol{\theta}$ and length selection embedding $\mathbf{W}$, we propose a training strategy with two phases towards better control of joint optimization

$$\begin{aligned} \boldsymbol{\theta} &\leftarrow \arg \min_{\boldsymbol{\theta}} \mathcal{L} + \lambda \mathcal{L}_{\text{dec}}, \\ \mathbf{W} &\leftarrow \arg \min_{\mathbf{W}} \mathcal{L} + \mu \mathcal{L}_{\text{Lap}}, \end{aligned} \tag{14}$$

where $\lambda > 0$ and $\mu > 0$ are hyperparameters. The two phases above alternate during model training.

## 4 EXPERIMENTS

In this section, we compare the performance of our approach with other methods on multiple benchmark datasets, conduct case studies to analyze the model's behavior, and investigate model variations in ablation studies.

### 4.1 DATASETS AND EVALUATION METRICS

We evaluate our method using the following five annotated real and synthetic datasets:

UCR[3]: the well-known subsequence anomaly dataset from "KDD cup 2021 multi-dataset time series anomaly detection" competition, consisting of 250 univariate time series from different domains with one subsequence anomaly per series. The lengths of the series vary from 4000 to one million.

UCR-Aug[4]: since UCR contains one anomaly per time series, we augment UCR by adding various types of subsequence anomalies with variable-length into each time series which is more consistent with most real-world scenarios.

SMAP and MSL[5]: Soil Moisture Active Passive satellite (SMAP) and Mars Science Laboratory rover (MSL), two datasets published by NASA (Hundman et al., 2018), with 55 and 27 series respectively. The lengths of the time series vary from 300 to 8500 observations.

SMD: Server Machine Dataset Su et al. (2019), a 5 weeks long dataset with 28 38-dimensional time series each collected from a different machine in large internet companies.

**Evaluation Metrics.** For the first two subsequence anomaly datasets, We choose AUC and Recall@$k$ as evaluation metrics, where AUC stands for the area under receiver operating characteristic (ROC) curves, and Recall@$k$ is the recall rate for the number of anomalous subsequences found in the top-$kn$ anomaly scored ones where $n$ is the total number of anomalous subsequences in one time series, i.e., for each anomaly, we have $k$ reporting opportunities considering the cost of check is limited in reality. For the last three multivariate datasets, we report F1 scores computed by choosing the best threshold on the test set following prior works.

---

[3]https://www.cs.ucr.edu/~eamonn/time_series_data_2018/

[4]This dataset is included in the Anonymous GitHub link in the Abstract.

[5]SMAP, MSL and the following SMD have a predefined train/test split, we do not use labels in training set following unsupervised paradigm.

Table 1: Performance of models on UCR.

| Metric | AUC | Recall@1 | Recall@3 | Recall@5 | Recall@10 |
|---|---|---|---|---|---|
| Matrix Profile | 0.8631 | 0.436 | 0.548 | 0.596 | 0.644 |
| Series2Graph | 0.6847 | 0.276 | 0.348 | 0.372 | 0.408 |
| LOF | 0.5995 | 0.312 | 0.360 | 0.409 | 0.461 |
| NCAD | 0.6413 | 0.352 | 0.412 | 0.520 | 0.648 |
| GraphSAD | **0.8864** | **0.444** | **0.572** | **0.640** | **0.696** |

Table 2: Performance of models on UCR-Aug.

| Metric | AUC | Recall@1 | Recall@3 | Recall@5 | Recall@10 |
|---|---|---|---|---|---|
| Matrix Profile | 0.8683 | 0.709 | 0.841 | 0.873 | 0.899 |
| Series2Graph | 0.7252 | 0.442 | 0.610 | 0.687 | 0.761 |
| LOF | 0.6129 | 0.594 | 0.736 | 0.798 | 0.861 |
| NCAD | 0.8414 | 0.722 | 0.803 | 0.839 | 0.890 |
| GraphSAD | **0.9022** | **0.792** | **0.925** | **0.953** | **0.981** |

## 4.2 IMPLEMENTATION DETAILS

**Subsequence split.** We set the length of an indivisible segment $\Delta = 0.125\ m$ for periodic time series with period length $m$, and $\Delta = 10$ for non-periodic time series. We utilize a sliding window with a stride $\tau = 2\Delta$ to generate subsequences. For the multi-scale length of subsequences, we set the maximum length scale $P = 5$, and thus the length can vary from 0.125 to 4 times of period length for periodic time series, which can stand for most common anomalies.

**Extension to multivariate data**. GraphSAD can be naturally extended to multivariate data as neural networks support multivariate input. We just need to modify the graph construction strategy. For the semantic graph, we calculate the distance for each variate and encode them into edge attributes, and for the temporal graph, we only retain the neighboring connection.

Due to space limitations, we only list these important experimental settings. For more details, e.g., hyperparameters, learning rate, and operational device, we refer to Appendix C.1.

## 4.3 COMPARISON EXPERIMENTS

We compare GraphSAD with detectors that are specified for subsequence anomaly detection used subsequence anomaly datasets, including Matrix Profile (Yeh et al., 2016) (a time series discord search method), Series2Graph (Boniol & Palpanas), LOF (Breunig et al., 2000), and NCAD (Carmona et al., 2022). Table 1 and 2 report the model performance on UCR and UCR-Aug, respectively. Our method outperforms Matrix Profile (the second best method) by a reasonable margin on UCR and a large margin on UCR-Aug. This is mainly because UCR has one anomaly per time series, to which time series discords are naturally applicable. While each time series in UCR-Aug can contain multiple anomalies and some of them are similar in anomalous patterns.

Table 3 shows the performance on multivariate datasets of GraphSAD compared against the state-of-the-art deep learning methods, including DeepSVDD (Ruff et al., 2018), OmniAnomaly (Su et al., 2019), THOC (Shen et al., 2020), and NCAD (Carmona et al., 2022). GraphSAD outperforms all baselines on two of the datasets, while its score is only slightly lower than NCAD on MSL. Considering that anomalies in these three datasets are mainly point-wise, which is not our main concern, yet GraphSAD still remains highly competitive.

Table 3: F1 score of models on multivariate datasets.

| Dataset | SMAP | MSL | SMD |
|---|---|---|---|
| Deep SVDD | 71.71 | 88.12 | – |
| OmniAnomaly | 84.34 | 89.89 | 88.57 |
| THOC | 95.18 | 93.67 | – |
| NCAD | 94.45 | **95.6** | 80.16 |
| GraphSAD | **96.67** | 95.32 | **91.12** |

Figure 5: GraphSAD Can Handle Recurring Anomalies Compared with Matrix Profile. There are 5 subsequence anomalies (marked in grey) in time series, where Anomaly 3 and 4 share a similar anomalous pattern. Matrix Profile find 1 of 5 anomalies in its top-5 scored subsequences which cannot deal with recurring anomalies, while GraphSAD find 4 of 5.

Table 4: Performance of model variations on UCR and UCR-Aug

| Dataset | UCR | | | UCR-Aug | | |
|---|---|---|---|---|---|---|
| Metric | AUC | Recall@1 | Recall@5 | AUC | Recall@1 | Recall@5 |
| w/o semantic graph | 0.7498 | 0.366 | 0.609 | 0.8513 | 0.714 | 0.881 |
| w/o temporal graph | 0.8832 | **0.446** | 0.587 | 0.8991 | 0.766 | 0.923 |
| w/o anomaly injection | 0.8640 | 0.440 | 0.587 | 0.9003 | 0.745 | 0.939 |
| w/o length selection | 0.8339 | 0.428 | 0.620 | 0.8731 | 0.762 | 0.936 |
| fixed length | 0.8765 | 0.396 | 0.594 | 0.8647 | 0.741 | 0.900 |
| GraphSAD | **0.8864** | 0.444 | **0.640** | **0.9022** | **0.792** | **0.953** |

## 4.4 CASE STUDIES

Here we conduct case studies to investigate how well GraphSAD handles recurring anomalies with similar patterns. Figure 5 illustrates a heartbeat time series in UCR-Aug. There are 5 subsequence anomalies (marked in grey) in time series, where Anomaly 3 and 4 share a similar anomalous pattern. Matrix Profile find 1 of 5 anomalies in its top-5 scored subsequences which cannot deal with recurring anomalies, while GraphSAD find 4 of 5. It can be seen that the proposed GraphSAD algorithm detects recurring anomalous subsequences much better than the Matrix Profile algorithm.

## 4.5 ABLATION STUDIES

To better understand the effectiveness of each component in GraphSAD, we perform ablation studies on UCR and UCR-Aug. We introduce model variations as in Table 4, where the first three are used to verify the effectiveness of semantic graph, temporal graph, and anomaly injection, the variation " w/o length selection" denotes multi-scale representations are averaged before input to GNN, i.e., different lengths are treated equally, and the variation "fixed length" denotes that we do not use the multi-scale representations but select a fixed subsequence length (which is set to be the period length for periodic time series and a constant 160 for the non-periodic). Results in Table 4 demonstrate the advantages of each component. It is important to note that without the semantic graph, the performance degrades a lot, suggesting the significance of discords-based prior knowledge.

## 5 CONCLUSION

We propose to bridge the gap between time series discords and deep learning-based TSAD methods and introduce a novel subsequence anomaly detection approach (GraphSAD). In addition to incorporating effective heuristics into neural networks, the model can also adaptively learn a proper subsequence length through a length selection mechanism, benefiting the subsequence AD problem. Experiments on multiple datasets demonstrate the effectiveness of the proposed model.

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

# A  RELATED WORK

Subsequence anomaly detection in time series remains a challenging problem in the time series community. In this section, we briefly introduce some state-of-the-art algorithms for subsequence time series anomaly detection, including discord based methods, one-class based methods, reconstruction based methods, and other methods.

## A.1  ANOMALY DETECTION BASED ON DISCORDS.

Subsequence anomaly detection can be formulated by finding discords in time series, where the discords are the subsequences maximally different from other subsequences in the same time series (Keogh et al., 2005). In Yankov et al. (2007), it is defined as the subsequence with the highest distance to its nearest neighbors. The distances can be calculated directly on the original signal (Yankov et al., 2007), or on the representations like wavelets (Fu et al., 2006). Many methods in this category have been proposed, like Matrix Profile (Yeh et al., 2016) and its latest variants (Nakamura et al., 2020; Lu et al., 2022). Matrix profile turns out to be an efficient solution for easy settings due to its efficient distance computation implementation. One limitation of matrix profile is the challenging parameter tuning. For example, its performance would degrade significantly if the window size is not properly set. In addition, it can easily capture the most different anomaly, but fails to detect similar anomalies which recur multiple times as illustrated in previous examples. Although this can be mitigated to some extent by taking $k$-th nearest neighbor into consideration (Yankov et al., 2008), it is hard to select an appropriate value of $k$ and the performance is highly sensitive to $k$.

## A.2  ANOMALY DETECTION BASED ON ONE-CLASS CLASSIFIER.

One-class approaches are common since the majority of the data is usually normal in anomaly detection. The One-Class Support-Vector-Machine ( Schölkopf et al. (2001), OC-SVM) and Support Vector Data Description ( Tax & Duin (2004), SVDD) are two well-known methods in this category, which use a hyperplane and hypersphere to separate the normal data from anomalous data in the kernel-induced feature space, respectively. These methods can also be extended to time series anomaly detection (Ma & Perkins, 2003). However, their performances are often not satisfied when facing complex and high-dimensional datasets, due to the curse of dimensionality and non-efficient computational scalability. To overcome these limitations, Deep Support Vector Data Description ( Ruff et al. (2018), DeepSVDD) is proposed to jointly train a deep neural network and optimize a data-enclosing hypersphere in output space. Inspired by Deep SVDD, the THOC (Shen et al., 2020) extends this deep one-class model by considering multiple spheres from all intermediate layers of a dilated recurrent neural network to extract multi-scale temporal features for better performance. Furthermore, Neural Contextual Anomaly Detection ( Carmona et al. (2022), NCAD) incorporates the idea of Hypersphere Classifier (HSC) which improves the Deep SVDD by utilizing known anomalies data under the (semi-)supervised setting (Ruff et al., 2020), as well as a window-based approach specified for time series anomaly detection. Another popular one-class classifier based method is the Deep Autoencoding Gaussian Mixture Model ( Zong et al. (2018), DAGMM) which integrates a deep autoencoder with the Gaussian mixture model (GMM) to generate a low-dimensional representation to capture the normal data pattern. However, directly applying DAGMM in time series may bring inferior performance since it does not properly model temporal dependencies.

## A.3  ANOMALY DETECTION BASED ON RECONSTRUCTION.

The reconstruction-based models usually learn a representation of the time series in latent space, reconstruct the signal from that representation, and conduct anomaly detection based on the reconstruction error. The long short-term memory based variational autoencoder (Park et al. (2018), LSTM-VAE) adopts serially connected LSTM and VAE layers to obtain the latent representation, estimates the expected distribution from the representation, and detects an anomaly when the log-likelihood of current observation given the expected distribution is lower than a threshold. Omni-Anomaly (Su et al., 2019) designs a stochastic recurrent neural network with state space models and normalizing flows for multivariate time series anomaly detection. AnoGAN (Schlegl et al., 2017) proposes a deep generative adversarial network (GAN) to model data distribution and estimate their probabilities in latent space for anomaly detection. The Multi-Scale Convolutional Recursive En-

coder Decoder ( Zhang et al. (2019), MSCRED) uses a convolutional encoder and convolutional LSTM network to capture the inter-series and temporal patterns, respectively, and adopts a convolutional decoder to reconstruct the input time series for anomaly detection.

### A.4 OTHER ANOMALY DETECTION METHODS

The rest of the anomaly detection methods roughly include density-based, transformer-based, and graph-based schemes. The Local Outlier Factor ( Breunig et al. (2000), LOF) is a classic density-estimation method that assigns each object a degree of being an outlier depending on how isolated the object is with respect to the surrounding neighborhood. Anomaly Transformer (Xu et al., 2021) proposes a new anomaly-attention mechanism to replace the original attention module and compute the association discrepancy, which can amplify the normal-abnormal distinguishability in time series under a minimax strategy to facilitate anomaly detection. Series2Graph (Boniol & Palpanas) transforms time series subsequences into a lower-dimensional space and constructs a directed cyclic graph, where the graph's edges represent the transitions between groups of subsequences and can be utilized for anomaly detection. MTAD-GAT (Zhao et al., 2020) designs two graph attention layers for learning the dependencies of time series in both temporal and feature dimensions, and then jointly optimizes a forecasting-based model and a reconstruction-based model for better anomaly detection results. While the existing graph-based methods achieve good results in some scenarios, they still have difficulty modeling and adapting to the challenging variable-length subsequences anomalies.

## B  DISTANCE-BASED ANOMALY DETECTION METHODS

Distance-based anomaly detection methods usually calculate the distance between a target sample (or subsequence) and its reference in explicit data space or latent representation space as the anomaly score. We now detail a prevailing part of them.

### B.1  TIME SERIES DISCORDS

Time series discords compute anomaly score of $\mathbf{X}_i$ as

$$s(\mathbf{X}_i) = \left\| \text{z-norm}(\mathbf{X}_i) - \text{z-norm}(\mathbf{X}_i^{(k-\text{NN})}) \right\|^2, \tag{15}$$

where $\text{z-norm}(\mathbf{x}) = (\mathbf{x} - \text{mean}(\mathbf{x})) / \text{std}(\mathbf{x})$ returns zero-normalized subsequence with $\text{mean}(\cdot)$ and $\text{std}(\cdot)$ standing for mean and standard deviation of input subsequence, and the corresponding reference $\mathbf{X}_i^{(k-\text{NN})}$ is the $k$-th nearest neighbour [6] of $\mathbf{X}_i$. Despite its effectiveness and wide usage, it has several limitations as we discussed in Section 1.

### B.2  DEEP SUPPORT VECTOR DATA DESCRIPTION BASED METHODS

The OC-SVM and SVDD rely on a proper kernel to map data features to a high dimensional space for data separation. And the DeepSVDD algorithm replaces the kernel-induced feature space in the SVDD method with the feature space learned in a deep neural network. Specifically, DeepSVDD (Ruff et al., 2018) is an unsupervised anomaly detection method that solves the following optimization problem

$$\min \frac{1}{N} \sum_{i=1}^{N} \|\text{NN}(\mathbf{X}_i) - \mathbf{c}\|^2, \tag{16}$$

where DeepSVDD calculates the latent distance $\|\text{NN}(\mathbf{X}_i) - \mathbf{c}\|^2$ as anomaly scores, $\text{NN}(\cdot)$ is a deep neural network, and $\mathbf{c}$ is a global reference which is the center of all the training data.

### B.3  HYPERSPHERE CLASSIFIER

The Hypersphere Classifier ( Ruff et al. (2020), HSC) improves DeepSVDD by training the network with the binary cross entropy objective function, which extends the model to training with labeled anomalies. In particular, the HSC loss is given by

$$-(1 - y_i) \log l(\text{NN}(\mathbf{X}_i)) - y_i \log(1 - l(\text{NN}(\mathbf{X}_i))), \tag{17}$$

---

[6] $k$ is usually set to 1.

where $y_i \in \{0, 1\}$ with 0 for the normal and 1 for the anomalous, and $l : \mathbb{R}^d \rightarrow (0, 1)$ maps the representation to a probabilistic prediction. Choosing $l(z) = \exp(-\|z\|^2)$ would reduce equation 17 to DeepSVDD objective with center $\mathbf{c} = 0$ when all labels are 0.

## C  EXPERIMENTAL DETAILS

### C.1  IMPLEMENTATION DETAILS OF GRAPHSAD

**Hyperparameters.** For hyperparameter tuning, we randomly select 8 time series to perform grid search. We use this inferred set of hyperparameters for all datasets. The hyperparameter settings are summarized in Table 5.

Table 5: Hyperparamter settings

| Hyperparameter | Description | Value |
|---|---|---|
| $\Delta$ | the length of an indivisible segment | 0.125 times of period length for periodic time series, and 10 otherwise |
| $\tau$ | stride length | $2\Delta$ |
| $P$ | the maximum length scale | 5 |
| $L$ | the maximum subsequence length | $2^P \Delta$ |
| $S$ | the number of hops when calculating contexts | 2 |
| $\alpha$ | the coefficient to balance semantic and temporal anomaly score | 0.8 |
| $\lambda$ | the coefficient of auto-encoding regularization | 1.0 |
| $\mu$ | the coefficient of Laplacian regularization | 0.2 |

**Model Training.** We run the model 5 times on each of the benchmark datasets and report the average score. We train the model with a Tesla V100 32GB GPU. For all datasets, we train Graph-SAD for 10 epochs with a learning rate of $1 \times 10^{-4}$ for model parameters and $5 \times 10^{-4}$ for length selection embeddings.

