# OpenReview forum: "Time Series Subsequence Anomaly Detection via Graph Neural Networks"
_ICLR.cc/2023/Conference — Submitted to ICLR 2023_

### Official Review · Reviewer_8d5u · 2022-10-22

**Confidence:** 3
**Correctness:** 3
**Technical Novelty And Significance:** 3
**Empirical Novelty And Significance:** 3
**Recommendation:** 5

**Clarity, Quality, Novelty And Reproducibility:**

Well written

Overall quality of work is good, experiments are convincing

The authors have released an anonymous link for code, indicating reproductibility

**Strength And Weaknesses:**

Strengths: Fairly well written and motivated well, appears technically novel from an application perspective, experiments seem convincing
Weakness: The key distinction from so many of the competing baselines is not argued well (e.g. Series2graph, THOC), which makes it slightly unclear what the new principle of this paper is compared to the literature, as these papers also try to address the same problem from a certain angle (e.g. subsequence, multi-scale).

**Summary Of The Paper:**

The paper addresses the problem of time-series subsequence anomaly detection, and this is executed through converting the time-series to a graph, and then performing anomaly detection on this graph representation. The problem is motivated by the fact that time-series anomalies can be subsequences, while most existing work focuses on point anomalies.

The method starts with splitting the time-series into multiple subsequences, followed by construction of graph priors - semantic priors and temporal priors. Semantic graphs is based on distances measures and incorporates multi-scale lengths. Temporal graph also uses multiple heuristics to incorporate temporal information. These graphs are followed by a multi-scale feature encoder and a length selection module to aid representation learning for the time-series. Auto encoding regularization and Laplacian regularization is used to stabilize the training. Experiments are performed on both synthetic and real world datasets, and GraphSAD achieves competitive performance - even performs well for point anomaly detection which is not the focus of the algorithm.

**Summary Of The Review:**

Overall the paper reads well and is a well done work (though not something I'd consider as breakthrough). I'll be happy to raise score after author's justify the key technical novelty compared the baselines addressing the same problem.

---

> ### Author Response · Authors · 2022-11-18
> **Response to Reviewer 8d5u**
>
> Thanks for your detailed and constructive feedbacks. We address your concerns and questions in the following.
>
> **Clarification:** As shown in ablation studies (Table 4) in the paper, **semantic graph** is the most important (the variant “w/o semantic graph” achieves the poorest performance).
>
> **Technical novelty:**  To the best of our knowledge, GraphSAD builds the graph to capture the temporal and distance information in a completely new way. GraphSAD first introduces semantic graphs of subsequences, which provide useful prior knowledge of subsequence distance for downstream anomaly detection (AD). In comparison, most existing works utilize graphs to describe relationships among different variables of multivariable time series. Others, e.g., Time2Graph[1] and Series2Graph[2], utilize graphs to represent state transition in time series, which are still quite different as they only consider temporal information (playing a similar role as temporal graph in GraphSAD, while periodic relationships are still not considered). As a result, GraphSAD is the first to encode subsequence distance into graph (i.e., semantic graph) to provide useful prior knowledge for downstream AD.
>
> Moreover, learning an appropriate subsequence length is quite significant for subsequence AD. Despite the proposed length selection module seems straightforward, it has not been studied previously and we attribute our contribution here to propose a new perspective (i.e., learning to select length) to innovate subsequence AD. While the candidate lengths are set multi-scaled, “Multi-scale” is not all-sided or exact here. Existing works (e.g., THOC [3]) learn a multi-scale representations for a certain subsequence, but in this work we aims at learning appropriate lengths from a initial subsequence for better dipicting its characteristics.
>
> Last but most important, the experimental results reveal GraphSAD outperforms other state-of-the-art algorithms on many benchmark datasets. We believe simple, effective, and easy-to-use algorithms are more preferred in practice and is our first priority.
>
> **Experiments:** We newly conduct experiments with more baselines, more real datasets, and other suitable metrics following your suggestions. The results are in general response above.
>
> [1] Ziqiang Cheng, et al. Time2graph: Revisiting time series modeling with dynamic shapelets. AAAI 2020.
> [2] Paul Boniol, et al. Series2Graph: Graph-based Subsequence Anomaly Detection for Time Series. VLDB 2020.
> [3] Lifeng Shen, et al. Time series anomaly detection using temporal hierarchical one-class network. NeurIPS 2020.

---

### Official Review · Reviewer_GTLu · 2022-10-23

**Confidence:** 4
**Correctness:** 3
**Technical Novelty And Significance:** 2
**Empirical Novelty And Significance:** 2
**Recommendation:** 6

**Clarity, Quality, Novelty And Reproducibility:**

Well-written paper; code is provided to assist with the reproducibility of results.

Novelty is somewhat low - nice engineering effort to put existing "practical knowledge" together, but the evaluation is lacking (no appropriate measures, no appropriate datasets, not all recent state-of-the-art classifiers).

**Strength And Weaknesses:**

Strengths:

1. Important and timely problem for time series
2. Well-written paper and easy to follow
3. Practical ideas fused appropriately into a single method

Weaknesses

1. Lack of novelty, mainly fusion of ideas
2. Missing several baselines
3. Missing several appropriate datasets and evaluation measures

Comments:

- The ideas are reasonable and appropriately merged into a single DNN/GNN framework. Unfortunately, there is a lack of technical depth and novelty. If we abstract the contributions, the core idea is to construct two types of graphs, and both such kinds of graphs exist/are well known (no novelty here). The remaining parts are mainly plug-ins of existing solutions. There is some interesting concepts in choosing lengths, but really similar solutions could be applied to every solution (e.g., run the method with different parameters, select on that maximizes anomaly score or using some voting etc.)

- Core very recent works are missing. There has been new works published well in-advance before the ICLR submission about new measures for evaluating subsequence anomaly detection methods as well as new benchmarks with 2000+ time series. Unfortunately, the paper does not consider the more recent advances in the area, and some methods are outdated. This is a fast pacing field, unfortunately.

"Tsb-uad: an end-to-end benchmark suite for univariate time-series anomaly detection." Proceedings of the VLDB Endowment 15.8 (2022): 1697-1711.

"Volume under the surface: a new accuracy evaluation measure for time-series anomaly detection." Proceedings of the VLDB Endowment 15.11 (2022): 2774-2787.

**Summary Of The Paper:**

The paper proposes GraphSAD, a subsequence time-series anomaly detection method that fuses discord-based anomaly detection solutions with graph neural network (GNN) /deep neural network (DNN) architectures. The core contribution of the paper is the construction of two types of graphs, semantic and temporal, to capture the distances and temporal dependencies in the data.  In addition, the proposed method proposes a solution for selecting the appropriate length for subsequences, a core parameter for that family of methods. Experimental results demonstrate the benefits of the proposed solution against several baselines on multiple datasets.

**Summary Of The Review:**

I believe the paper is on the correct track to getting accepted eventually in a top venue. However, I have significant concerns about the evaluation of the work: missing recent comprehensive benchmarks in the area and missing the appropriate methods to evaluate subsequence methods. For the baselines, similar tricks for selecting the length should be used to ensure fairness.

From the ablation study, we see that the most important "idea" is the temporal graph. Semantic graph lacks behind, which contradicts the entire concept of using GNNs in the first place. Temporal information seems sufficient and tons of methods exist (and not being evaluated) to capture these cases. Check references for appropriate evaluation on new benchmarks and new evaluation measures, using the appropriate baselines.

---

> ### Author Response · Authors · 2022-11-18
> **Response to Reviewer GTLu**
>
> Thanks for your detailed and constructive feedbacks. We address your concerns and questions in the following.
>
> To the best of our knowledge, GraphSAD first introduces semantic graphs providing useful prior knowledge of subsequence distance for downstream anomaly detection (AD). It builds the graph to capture the temporal and distance information in a completely new way. Most existing works utilize graphs to describe relationships among different variables of multivariable time series. Others, e.g., Time2Graph[1] and Series2Graph[2], utilize graphs to represent state transition in time series, which are still quite different as they only consider temporal information (playing a similar role as temporal graph in GraphSAD, while periodic relationships are not considered). Moreover, learning an appropriate subsequence length is crucial but challenging for accurate subsequence AD. A potential method is to perform AD with multiple lengths and vote for final anomaly scores. However, results of different lengths cannot be aligned, and we find that this strategy commonly failed in our try outs. Despite the proposed module of learning to select length seems straightforward, it has not been studied previously and we attribute our contribution here to propose a new perspective (i.e., learning to select length) to innovate subsequence AD.
>
> Moreover, we newly conduct experiments with more baselines, more real datasets, and other suitable metrics following your suggestions. The results are in general response above.
>
> Last but most important, the experimental results reveal GraphSAD outperforms other state-of-the-art algorithms on many benchmark datasets. We believe simple, effective, and easy-to-use algorithms are more preferred in practice and is our first priority.
>
> [1] Ziqiang Cheng, et al. Time2graph: Revisiting time series modeling with dynamic shapelets. AAAI 2020.
> [2] Paul Boniol, et al. Series2Graph: Graph-based Subsequence Anomaly Detection for Time Series. VLDB 2020.

---

> > ### Comment · Reviewer_GTLu · 2022-11-22
> > **thanks for addressing my comments**
> >
> > The updated experiments have significantly improved the clarity of your results (more SOTA baselines, appropriate evaluation measures + datasets). Therefore, I am happy to increase by 2 scales my score (reject -> weak accept).
> >
> > You can focus your attention on addressing the comments of the other reviewers :)

---

### Official Review · Reviewer_s3os · 2022-10-24

**Confidence:** 4
**Clarity, Quality, Novelty And Reproducibility:** See Strength and Weakness.
**Correctness:** 2
**Technical Novelty And Significance:** 2
**Empirical Novelty And Significance:** 2
**Recommendation:** 3

**Strength And Weaknesses:**

Strength
1. The proposed method considers two versions of graphs to encode the relationship between subsequences, a distance based graph and periodic pattern based graph.
2. The method introduces a subsequence length selection mechanism which is trained together with other parts of the model so that the length selection problem is mitigated to some extent.
3. The experimental results demonstrate the effectiveness of the proposed method in terms of improvements over several baseline methods.

Weakness
1. The overall novelty of the proposed method is limited. Using graph and multi-scale subsequence length of anomaly detection in time series has been studied before (e.g., A deep neural network for unsupervised anomaly detection and diagnosis in multivariate time series data. AAAI, 2019). Learning to select lengths is not significant innovation as it can be done by adding some attention based mechanisms. The paper briefly discussed these existing methods in the appendix, without sufficient details. Comparison may be necessary. The related work section is important, and should be moved to the main paper.
2. The proposed method may be limited by the initial subsequence length selection, which was assumed to be large, without a clear selection guidelines and impacts on performance.
3. It is not clear what the dimension $d_{e}$ is in $E_{j,i}^{(se)}$. It seems to be 1 for representing the semantic proximity.
4. The temporal graph only encodes periodic patterns, which may ignore other temporal dependency patterns.
5. The meaning of $Z_{i,0}, ..., Z_{i,P}$ are not clear. The question is why to aggregate information for all lengths before $l$, i.e. , $R_{i, :l}$, instead of using $l$ directly, i.e., $R_{i,l}$.
6. The intuition behind the length selection method Eq 4 and Eq 5 should be better elaborated. The current description is unclear because $Z_{i,P}$ represents aggregation for all lengths less than $l$, instead of using $l$ directly.
7. The motivation to use GNN to encode context information for each subsequence representation is also unclear. To detect anomalies, the method aims to find the difference between a subsequence and its context using Eq 9. GNN seems do the contrary. It aims to mitigate the difference between a subsequence representation and its context.
8. The authors may want to discuss if the model trained with injected anomalies is biased to detect anomaly patterns follows the injection. This could be further investigated in the experiments. Also Eq. 11 does not consider unbalanced classes (normal and abnormal).
9. The experiments could be further improved by including more evaluation metrics, and baseline methods. There are more subsequence based anomaly detection methods, as discussed in Anomaly Detection in Time Series: A Comprehensive Evaluation, VLDB 2022. Some of them could be evaluated in the experiments.


**Summary Of The Paper:**

This paper presents a method for anomaly detection in time series. The proposed method constructs two graphs of subsequences of the input time series, by encoding subsequence distance and periodic patterns, with multiple choices of length scales of subsequences. The model uses GNN and length embeddings to learn to weigh different scales. The aim is to capture resolve the problem of selecting subsequence lengths. The experimental results demonstrate the method outperforms some baseline methods.

**Summary Of The Review:**

The paper introduces a graph based subsequence anomaly detection in time series. The overall novelty of the proposed method is limited. The method design and experiments remain to be further justified and improved.

---

> ### Author Response · Authors · 2022-11-18
> **Part 1 of Response to Reviewer s3os**
>
> Thanks for your detailed and constructive feedbacks. We address your concerns and questions in the following.
>
> **Q1.** The overall novelty of the proposed method is limited. Using graph and multi-scale subsequence length of anomaly detection in time series has been studied before. Learning to select lengths is not significant innovation as it can be done by adding some attention based mechanisms.
>
> **R1.** Most existing works introduce graphs to describe relationships among different variables of multivariable time series, while the graphs we used in this paper is to depict semantic and temporal relationships of extracted subsequences. To the best of our knowledge, GraphSAD builds the graph to capture the temporal and distance information in a completely new way. Existing works  Time2Graph[1] and Series2Graph[2] utilize graphs to represent state transition in time series for anomaly detection (AD), which are still quite different as they only consider temporal information (playing a similar role as temporal graph in GraphSAD, while periodic relationships are not considered). In fact, GraphSAD is the first to encode subsequence distance into graph (i.e., semantic graph) to provide useful prior knowledge for downstream AD. Moreover, learning an appropriate subsequence length is crucial but challenging for accurate subsequence AD. Despite the proposed strategy of learning to select lengths seems straightforward, it has not been proposed and studied previously and we attribute our contribution here to propose a new perspective (i.e., learning to select length) to innovate subsequence AD. Last but most important, the experimental results reveal GraphSAD outperforms other state-of-the-art algorithms on many benchmark datasets. We believe simple, effective, and easy-to-use algorithms are more preferred in practice and is our first priority.
>
> **Q2.** The proposed method may be limited by the initial subsequence length selection, which was assumed to be large, without a clear selection guidelines and impacts on performance.
>
> **R2.** Actually, we choose 4 times of period length as the largest subsequence length, which is enough to support detecting various anomalies. Even though some anomaly subsequence exceed this length, it can still depict the characteristics of such anomaly. For non-periodic time series, one should determine the initial maximal length according to the data source, sampling frequency, etc., where there is no clear criterion. In our experiments, we set this length to 800 (in datasets SMAP, MSL, and SMD).
>
> **Q3.** It is not clear what the dimension de is in Ej,i(se). It seems to be 1 for representing the semantic proximity.
>
> **R3.** We use Euclidean distance and z-norm Euclidean distance as measures, as well as we calculate distance for different lengths (i.e., $l = 2^p \Delta, p = 0,1,\cdots,P$, and $\Delta$ is the length of an indivisible segment). As a result, the edge attribute in semantic graph is multi-dimensional, and the dimension $d_e = 2(P+1)$.
>
> **Q4.** The temporal graph only encodes periodic patterns, which may ignore other temporal dependency patterns.
>
> **R4.** Actually, we encode periodic and adjacent relationships in temporal graph, which are the most important temporal relationships for time series data.
>
> **Q5.**	The meaning of Zi,0,...,Zi,P are not clear. The question is why to aggregate information for all lengths before l, i.e. , Ri,:l, instead of using l directly, i.e., Ri,l.
>
> **R5.** As $ R_i $ is generated from $ X_i $ using causal convolution network with limited receptive field, each element $ R_{i,l} $ cannot include all information of subsequence $ X_{i, 1:l} $ (or $ X_{i, :l} $). So we introduce statistics pooling to aggregate $ R_{i,:l} $ to represent subsequence of length $l$ (i.e., from time step 1 to $l$).
>
> **Q6.**	The intuition behind the length selection method Eq 4 and Eq 5 should be better elaborated. The current description is unclear because Zi,P represents aggregation for all lengths less than l, instead of using l directly.
>
> **R6.** Please refer to Response 5.
>
> **Q7.**	The motivation to use GNN to encode context information for each subsequence representation is also unclear. To detect anomalies, the method aims to find the difference between a subsequence and its context using Eq 9. GNN seems do the contrary. It aims to mitigate the difference between a subsequence representation and its context.
>
> **R7.** GNN not only aggregates the neighboring subsequences but the corresponding edge attributes (i.e., semantic and temporal proximities) as messages. Besides, the message function (eq.1) is implemented with nonlinear MLP, which provide enough flexibility of utilizing the neighboring information. For instance, it can absorb large amount of information from close neighbor (with large edge attributes) as both of them tend to be normal or abnormal simultaneously but little from less similar neighbors.

---

> > ### Author Response · Authors · 2022-11-18
> > **Part 2 of Response to Reviewer s3os**
> >
> > **Q8.** The authors may want to discuss if the model trained with injected anomalies is biased to detect anomaly patterns follows the injection. This could be further investigated in the experiments. Also Eq. 11 does not consider unbalanced classes (normal and abnormal).
> >
> > **R8.** Although normal and anomalous data are significantly imbalanced in anomaly detection task, it is different from imbalanced classification, as anomalous data have various patterns and cannot be grouped into one category. As a result, many anomaly detection methods [3,4,5] aim at learning compact normal representations, compared with which anomalies are detected, and labeled anomalies are not necessary to train a model (namely one-class classification). Eq.11 is derived from HSC loss[6], and without labeled anomalies, it degrades to one-class classification loss. Moreover, recent works [4,6] suggest that with several injected outliers (even though they have no consistent characteristics with real anomalies), the model can learn more expressive normal representations. So the imbalanced nature is not a issue here.
> >
> > **Q9.**	The experiments could be further improved by including more evaluation metrics, and baseline methods. There are more subsequence based anomaly detection methods, as discussed in Anomaly Detection in Time Series: A Comprehensive Evaluation, VLDB 2022. Some of them could be evaluated in the experiments.
> >
> > **R9.** We newly conduct experiments with more baselines, more real datasets, and other suitable metrics following your suggestions. The results are in general response above.
> >
> > [1] Ziqiang Cheng, et al. Time2graph: Revisiting time series modeling with dynamic shapelets. AAAI 2020.
> > [2] Paul Boniol, et al. Series2Graph: Graph-based Subsequence Anomaly Detection for Time Series. VLDB 2020.
> > [3] Lukas Ruff, et al. Deep one-class classification. ICML 2018.
> > [4] Chris U. Carmona, et al. Neural Contextual Anomaly Detection for Time Series. IJCAL 2022.
> > [5] Lifeng Shen, et al. Time series anomaly detection using temporal hierarchical one-class network. NeurIPS 2020.
> > [6] Lukas Ruff, et al. Rethinking assumptions in deep anomaly detection.

---

> > ### Comment · Reviewer_s3os · 2022-12-05
> > **Thanks for the response**
> >
> > Thank the authors for the response. From the paper and the author response, I am not fully convinced about the significance of the work. The response describes the existing methods used graphs to encode the relationships between variables, and the proposed method used graphs to encode semantic and temporal relationships. The method in (Anomaly transformer: time series anomaly detection with association discrepancy. In ICLR, 2022) used the attention mechanism in transformer to encode the temporal relationships, i.e., the S matrix in Equation 2. Although it didn't use GNNs, but S could also be considered as a graph of temporal relationships. It also provides a way to automatically learn the width of the subsequence areas around each time step. As for multi-scale subsequences, the method I mentioned in the original Q1, i.e., (A deep neural network for unsupervised anomaly detection and diagnosis in multivariate time series data. AAAI, 2019), has explored it. The author response ignored this method. It is better to elaborate the differences between the proposed method and these methods, and make the advantages from the differences clear, instead of describing the differences. For example, it is not clear what are the impacts from the different ways of modeling the temporal relationships between the proposed method and the anomaly transformer. The related work discussion in the current paper remains to improve. I still think the related work section should be included in the main paper rather than in the appendix considering its importance. Also, the compared methods in the experiments are insufficient for justifying the advantages of the proposed method. As suggested in my original questions, it is better to improve the experiments by including the aforementioned methods in the comparison for demonstrating the impacts of the different ways of modeling the relationships. The revised paper seems have the same compared methods as the previous version.

---

### Official Review · Reviewer_s2pX · 2022-11-29

**Confidence:** 4
**Correctness:** 3
**Technical Novelty And Significance:** 2
**Empirical Novelty And Significance:** 3
**Recommendation:** 5

**Clarity, Quality, Novelty And Reproducibility:**

I think the paper can be improved in terms of clarity and organization. The authors do a good job of explaining individual components, but the poor organization means that it is hard to understand the overall algorithm. The paper has limited originality. As I mentioned earlier, there is some innovation in using two types of graph representation. Otherwise the authors have strung up different components without much discussion on the justification for each of those design choices.

**Strength And Weaknesses:**

Strengths:
- The problem is important and has real-world applications.
- The idea of not relying on a single window length is good and could make the method applicable in more settings.

Weaknesses:
- The paper is not very well-written and it is hard to understand how all the steps fit together. While the overview figure is useful, it does not provide the complete picture.
- The paper seems to be putting together several individual components to solve the problem but is not necessarily pushing the boundary significantly. While I agree that the results are better than others, the comparison is not comprehensive enough to show that this will generalize well. I can see some novelty in using two different representations (semantic and temporal), but the authors do not discuss the impact of using both representations.
- The experimental results are not comprehensive. I would have liked to see comparisons with the vast literature on discord detection which have been shown to be very effective in such problems, and for many of the data sets used in this paper.
- While I commend the relaxation of the need to specify a window length, it is unclear what is impact of the choice of the initial window length on the overall performance

**Summary Of The Paper:**

The paper proposes a method to identify anomalous subsequences in a long time series using a graph neural network. The core idea is to use two different graph representations for fixed length subsequences from the time series, one which captures the semantic distance (based on different distance measures) and the second which captures the temporal distance (both time adjacency and periodicity). The method allows for capturing similarity between subsequences at different length scales. The two graph neural networks are then combined within a single learning framework to learn representations for the subsequences, which are then used for anomaly detection.

The method is compared to a few other methods and shown to outperform them on a selection of data sets.

**Summary Of The Review:**

Overall, the paper is targeting an important problem. There is some marginal benefit shown on a set of data sets over a selection of methods. But the results are not convincing enough to show that the approach is generally applicable. There are some questions on the novelty.

---

> ### Author Response · Authors · 2022-12-01
> **Part 1 of Response to Reviewer s2pX**
>
> Thanks for your detailed and constructive feedbacks. We address your concerns and questions in the following.
>
> **Technical contributions and remarks.**
>
> We summarize our main contributions as follows:
> 1. We propose to encode pratical heuristics of subsequence anomaly detection (AD) including distance and temporal relationships into graphs and appropriately exploit them with GNNs and multi-context aware hypersphere classification objective.
> 2. As learning an appropriate subsequence length is crucial but challenging for accurate AD, we propose a length selection mechanism which has not been proposed and studied previously and provides a new perspective to innovate subsequence AD.
> 3. We verify the effectiveness and generality of our proposed GraphSAD with extensive experiments. Newly conducted experiments on more real-world datasets with more baselines, and other suitable metrics can be found in "General Response".
>
> Last but most important, although the method seems straightforward, the experimental results reveal GraphSAD outperforms other state-of-the-art algorithms on many benchmark datasets. We believe simple, effective, and easy-to-use algorithms are more preferred in practice and is our first priority.
>
> **Response to questions.**
>
> **Q1.** The paper is not very well-written and does not provide the complete picture.
>
> **R1.** The overview of GraphSAD is provided in Figure 3 in the paper, where time series are first split into subsequences of initial (largest) length using sliding window, and both semantic and temporal graphs are constructed. Afterwards, a multi-scale (length) feature extractor and length selection mechanism are utlized to learn representations of each subsequence with proper length, which are fed into GNNs to incorporate neighbor and edge information for final anomaly detection.
>
> **Q2.** The comparison is not comprehensive enough to show that this will generalize well.
>
> **R2.** We newly conduct experiments with more baselines and other suitable metrics on more real-world datasets, as aknowleged by Reviewer GTLu. The results are in "General Response" which further verify the effectiveness and generality of GraphSAD.
>
> **Q3.** The authors do not discuss the impact of using semantic and temporal representations.
>
> **R3.** Previous works has partially studied semantic and temporal informantion for time series AD. Time series discords [1] and local outlier factor[2] (LOF) utilize semantic (or distance) information for AD, while they cannot take into account multiple distance measures at the same time and fall into drawbacks of model expressiveness. NCAD[3], Time2Graph[4], and Series2Graph[5] utilize temporal dependencies for AD, while they only consider the temporal adjacency. To the best of our knowledge, GraphSAD is the first to investigate semantic and temporal information systematically for AD. Besides, in our ablation studies, we investigate the effectiveness of both semantic and temporal representations empirically (Table 4).
>
> **Q4.** The experimental results are not comprehensive. I would have liked to see comparisons with the vast literature on discord detection which have been shown to be very effective in such problems, and for many of the data sets used in this paper.
>
> **R4.** Matrix Profile (MP) detects the exact discords (which are involed in our experiments), while most variants including Merlin[6] and HotSax[7] mainly focus on accelerating detection process with approximation methods which cannot outperform vanilla MP in detection accuracy. Thus, we report MP's performance as a representative of discord detection methods.

---

> > ### Author Response · Authors · 2022-12-01
> > **Part 2 of Response to Reviewer s2pX**
> >
> > **Q5.** It is unclear what is impact of the choice of the initial window length on the overall performance.
> >
> > **R5.** Actually, we choose 4 times of period length as the intial (largest) subsequence length that is enough to support detecting various anomalies. Even though the anomaly subsequence exceed this length, it can still depict the characteristics of such anomaly. For non-periodic time series, one should determine the initial maximal length according to the data source, sampling frequency, etc., where there is no clear criterion. In our experiments, we set this length to 800 (in dataset SMAP, MSL, and SMD).
> > Following your suggestions, we conduct additional sensitivity analysis experiments of the initial length. As shown in the following table, for dataset UCR (including subseqence anomaly with different lengths), 4 times of period length as initialization is the best as it can cover most abnormal patterns and does not lead to too much search space, and for dataset SMD (which has no obvious periodicy and most anomalies are point-wise), a small length (200) is enough. However, in data-agnoistic scenarios, 4 times of a segment length (200) is a proper choice.
> >
> > |length/Datasets |  UCR(AUC)   |SMD(F1) |
> > |     :----:     | :----: | :----: |
> > | 1 time of period/200 |   0.8372| **0.957**  |
> > | 2 times of period/400|   0.8667|    0.949   |
> > | 4 times of period/800|**0.8864**|   0.953   |
> > |6 times of period/1200 |   0.8812|   0.952   |
> > |8 times of period/1600 |   0.8843|   0.949   |
> > |12 times of period/2400|   0.8350|   0.942   |
> > |16 times of period/3200|  0.8119 |   0.936   |
> >
> >
> > [1] Chin-Chia Michael Yeh, et al. Matrix profile I: all pairs similarity joins for time series: a unifying view that includes motifs, discords and shapelets. ICDM 2016.
> > [2] Markus M Breunig, et al. LOF: identifying density-based local outliers. SIGMOD 2000.
> > [3] Lifeng Shen, et al. Time series anomaly detection using temporal hierarchical one-class network. NeurIPS 2020.
> > [4] Ziqiang Cheng, et al. Time2graph: Revisiting time series modeling with dynamic shapelets. AAAI 2020.
> > [5] Paul Boniol, et al. Series2Graph: Graph-based Subsequence Anomaly Detection for Time Series. VLDB 2020.
> > [6] Takaaki Nakamura, et al. MERLIN: Parameter-Free Discovery of Arbitrary Length Anomalies in Massive Time Series Archives. ICDM 2020.
> > [7] Eamonn Keogh, et al. HOT SAX: Finding the Most Unusual Time Series Subsequence: Algorithms and Applications. ICDM 2005.

---

### Author Response · Authors · 2022-11-18
**General Response for Commen Questions**

We conduct experiments with more baselines, more real datasets, and other suitable metrics following reviewers' suggestions.

**Metrics:**
We newly choose Range-AUC-ROC (R-AUC) and Volume Under the Surface of ROC (VUS-ROC, and VUS for simplicity)[1] as metrics particularly designed for subsequence anomaly detection considering buffer regions at the boundary of anomalies. We set the (maximal) boundary length to be the period length for periodic time series and a fixed length of 200 for non-periodic ones.

**Datasets:**
* $\mathrm{UCR}$ and $\mathrm{UCR\mbox{-}Aug}$ are involved in the paper, and we additionally add R-AUC and VUS as metrics.

The following real-world subsequence anomaly detection datasets are provided in [2] which we consider as new benchmark datasets.

* $\mathrm{SED}$, $\mathrm{ECG}$ and $\mathrm{IOPS}$ are collected from various domains with different characteristics.
* $\mathrm{MGAB}$, $\mathrm{NAB}$ and $\mathrm{SensorScope}$ are also provided in [2], however, the qualities of these datasets are pretty limited and their anomalies labels are not making any sense.

**Baselines:**
Besides methods including MP, Series2Graph, NCAD, etc. reported in the original paper, we add PCA, AE, NormA, IForest, LSTM-AD as baselines.
* $\mathrm{PCA}$: Principal component analysis (PCA) can be used in detecting outliers. PCA is a linear dimensionality reduction using Singular Value Decomposition of the data to project it to a lower dimensional space. In this procedure, covariance matrix of the data can be decomposed to orthogonal vectors, called eigenvectors, associated with eigenvalues. The eigenvectors with high eigenvalues capture most of the variance in the data. Therefore, a low dimensional hyperplane constructed by k eigenvectors can capture most of the variance in the data. However, outliers are different from normal data points, which is more obvious on the hyperplane constructed by the eigenvectors with small eigenvalues. Therefore, outlier scores can be obtained as the sum of the projected distance of a sample on all eigenvectors.
* $\mathrm{AutoEncoder}$ [3]: classical deep learning method.
* $\mathrm{NormA}$ [4]: the SOTA unsupervised learning method for subsequence anomaly detection in data mining.
*  $\mathrm{IForest}$: Isolation Forest (IForest) assumes a correlation between the distance of tree structure and the likelihood of being an anomly. Obviously, random partitioning produces noticeably shorter paths for anomalies. Hence, by randomly creating tree structure on the dataset, we may calculate the expected distance of each point and thus a measure of its normality.
* $\mathrm{LSTM\mbox{-}AD}$[5]: as a forecasting method, it learns normal patterns and heavily relies on seasonality/periodicity; for this reason, it easily identifies the anomalous subsequence, but since it is robust to noise, it ignores the point anomaly.

**Results:**
Results in Table 1 verify that GraphSAD outperforms other methods signicantly. Moverover, we find that GraphSAD consistently achives high performance on all datasets with the average ranking of 1.3/6, despite data are collected from different domains and have various characteristics, which demonstrates that GraphSAD is a general and powerful subsequence anomaly detection method.

**Table 1:** Performance of models on multiple datasets. We choose R-AUC (the left measure) and VUS (the right measure) as metrics, and the last column reports the average ranking of different models on multiple datasets.

|Methods/Datasets|  UCR   |UCR-Aug |  SED   |   ECG  |  IOPS  |AvgRank |
|     :----:     | :----: | :----: | :----: | :----: | :----: | :----: |
|       PCA      | 0.568/0.561|0.571/0.562| 0.587/0.537|0.756/0.754| 0.737/0.728|5.0|
|       AE       |0.770/0.763|0.861/0.848|0.962/0.951|0.596/0.584|0.686/0.679|4.2|
|      NormA     | 0.874/0.872 |0.942/0.935| **0.998**/ **0.994** |0.982/**0.978**|0.576/0.569|2.5|
|      IForest   |0.672/0.668|0.788/0.775| 0.669/0.650|0.976/0.955| 0.843/0.829 |3.8|
|    LSTM-AD     |0.676/0.672|0.767/0.760| 0.712/0.694|0.507/0.494| 0.651/0.649|4.8|
|    GraphSAD    |**0.908**/**0.899**|**0.960**/**0.945**|0.986/0.981|**0.985**/0.967|**0.911**/**0.904** |**1.3**|

[1] John Paparrizos, et al. Volume Under the Surface: A New Accuracy Evaluation Measure for Time-Series Anomaly Detection. VLDB 2022.
[2] John Paparrizos, et al. TSB-UAD: An End-to-End Benchmark Suite for Univariate Time-Series Anomaly Detection. VLDB 2022.
[3] Mayu Sakurada, et al. Anomaly Detection Using Autoencoders with Nonlinear Dimensionality Reduction. MLSDA 2014.
[4] Paul Boniol, et al. Unsupervised and Scalable Subsequence Anomaly Detection in Large Data Series. VLDBJ 2021.
[5] Pankaj Malhotra, et al. Long Short Term Memory Networks for Anomaly Detection in Time Series. ESANN 2015.

---

### Decision · Program_Chairs · 2023-01-20

**Decision:**

Reject

**Justification For Why Not Higher Score:**

After rebuttals, one referee raised their score, but it seems there is still insufficient enthusiasm from the referees to accept the current version of this paper.

**Justification For Why Not Lower Score:**

N/A

**Metareview: Summary, Strengths And Weaknesses:**

This paper studied an important problem motivated by real-world applications: identifying anomalous subsequences in a long time series using a graph neural network. The referees raised many concerns about the writing, limited novelty, contributions, insufficient experiments, and limitation by the initial subsequence length selection. After rebuttals, one referee raised their score, but it seems there is still insufficient enthusiasm from the referees to accept the current version of this paper at the 2023 ICLR.